Expression patterns of cysteine peptidase genes across the Tribolium castaneum life cycle provide clues to biological function

Perkin Lindsey 1 lindsey.perkin@ars.usda.gov
Elpidina Elena N. 2
Oppert Brenda 1
1 Center for Grain and Animal Health Research, USDA, Agricultural Research Service , Manhattan, KS , United States
2 A. N. Belozersky Institute of Physico-Chemical Biology, Moscow State University , Moscow , Russia
Li Jingyi Jessica
Electronic publication date: 2016 Jan 18
Publication date: 2016
Volume: 4
Electronic Location ID: e1581
Received 2015 Sep 8; Accepted 2015 Dec 18
Copyright year: 2016
License: This is an open access article, free of all copyright, made available under the Creative Commons Public Domain Dedication. This work may be freely reproduced, distributed, transmitted, modified, built upon, or otherwise used by anyone for any lawful purpose.
License URL: https://creativecommons.org/publicdomain/zero/1.0/

Keywords: Transcriptome, Tribolium castaneum, Tenebrionid digestion, Gene expression across life stages, Cysteine peptidase

Funding: United States Department of Agriculture Agricultural Research Service CRIS 032 Russian Foundation for Basic Research 15-04-08689-a The authors received funding from the United States Department of Agriculture, Agricultural Research Service CRIS project 032. This research was also supported by Russian Foundation for Basic Research grant # 15-04-08689-a. The funders had no role in study design, data collection and analysis, decision to publish, or preparation of the manuscript.

==============================
The red flour beetle, Tribolium castaneum, is a major agricultural pest responsible for considerable loss of stored grain and cereal products worldwide. T. castaneum larvae have a highly compartmentalized gut, with cysteine peptidases mostly in the acidic anterior part of the midgut that are critical to the early stages of food digestion. In previous studies, we described 26 putative cysteine peptidase genes in T. castaneum (types B, L, O, F, and K) located mostly on chromosomes 3, 7, 8, and 10. In the present study, we hypothesized that specific cysteine peptidase genes could be associated with digestive functions for food processing based on comparison of gene expression profiles in different developmental stages, feeding and non-feeding. RNA-Seq was used to determine the relative expression of cysteine peptidase genes among four major developmental stages (egg, larvae, pupae, and adult) of T. castaneum. We also compared cysteine peptidase genes in T. castaneum to those in other model insects and coleopteran pests. By combining transcriptome expression, phylogenetic comparisons, response to dietary inhibitors, and other existing data, we identified key cysteine peptidases that T. castaneum larvae and adults use for food digestion, and thus new potential targets for biologically-based control products.

Introduction

Tribolium castaneum, the red flour beetle, is a highly destructive stored product pest, and cases of resistance to most control products have been identified (Mann, Adam & Arthur, 2013). The T. castaneum genome has been sequenced (Tribolium Genome Sequencing Consortium et al., 2008), providing a convenient model system to develop and evaluate alternative control strategies for coleopteran storage pests based on genetic analysis. Our previous biochemical and genetic studies have focused on the T. castaneum larval gut, as it is one of the main interfaces between the beetle and the environment. The larval gut of T. castaneum is compartmentalized, with an acidic anterior portion that secretes a high concentration of C1 family cysteine peptidases, while the slightly alkaline posterior portion of the midgut has mostly serine peptidases (Prabhakar et al., 2007; Vinokurov et al., 2009). The presence of cysteine peptidases in the coleopteran gut has been proposed as an adaptation to avoid serine peptidase inhibitors in grain kernels (Terra & Ferreira, 1994; Terra & Cristofoletti, 1996; Vinokurov et al., 2009). Cysteine peptidases also provide efficient digestion of cereal grain proteins (Goptar et al., 2012). T. castaneum larvae respond to cysteine peptidase inhibitors through a complex response, increasing the transcript expression of genes encoding specific serine and cysteine peptidases (Oppert et al., 1993; Oppert et al., 2003; Oppert et al., 2005; Oppert et al., 2010).

Cysteine peptidases from the C1 family are mostly found in lysosomes/endosomes in other organisms, but also can be located in other cellular compartments (Turk et al., 2012). Cysteine peptidases are active and stable at slightly acidic pH and are mostly irreversibly inactivated at neutral pH, which serves as a regulator of activity (Turk et al., 1995). In lysosomes, cysteine peptidases and other hydrolases degrade proteins, activate granule proteases, and participate in cellular processes like antigen presentation (Turk et al., 2012). However, cysteine peptidases also may be found extracellularly in a variety of tissues, and are involved in many biological processes, such as bone remodeling, keratinocyte differentiation, and prohormone activation. Cysteine peptidases have been implicated in human diseases, like cancer, arthritis, cardiovascular disease and others (Repnik et al., 2012). Activation of cysteine peptidase zymogens occurs when the proenzyme enters an intracellular compartment, such as the lysosome, or is released into an acidic extracellular environment. In vitro, the released propeptide domain can inhibit the activity of the mature enzyme (Coulombe et al., 1996), but cysteine peptidases mostly are regulated by their environment in vitro (i.e., pH) or endogenous inhibitors, such as cystatins, thyropins, and others (Turk et al., 2012).

In the T. castaneum genome, 25 cysteine peptidase genes and one associated pseudogene have been identified (Tribolium Genome Sequencing Consortium et al., 2008; Martynov et al., 2015), more than three times the number of cysteine peptidase genes in Drosophila melanogaster and other current model insects. T. castaneum cysteine peptidase genes include cathepsin B and L peptidases, mostly arranged in clusters on chromosome 3, 7, 8 and 10, and also single genes encoding cathepsins F, K, and O on chromosome 7, 4 and 1(X), respectively. Empirical evidence of the biological function of each cysteine peptidase gene in T. castaneum is lacking, although our previous studies suggest that some function as major processors of food proteins in the larval gut (Oppert et al., 1993; Vinokurov et al., 2009; Oppert et al., 2010; Martynov et al., 2015). Gene expression studies indicated that cathepsin L genes LOC659441 and LOC659502 are the most highly expressed cysteine peptidases in the larval gut and are most likely encoding enzymes important in the early stages of cereal protein digestion, as they are located in the anterior midgut (Prabhakar et al., 2007; Morris et al., 2009; Vinokurov et al., 2009; Martynov et al., 2015).

In the current study, we coupled existing knowledge of T. castaneum larval gut cysteine peptidases with new RNA-seq data from the four major life stages. We hypothesized that cysteine peptidase genes highly expressed during feeding stages (adults and larvae) are primarily for food digestion, especially those that coincide with high expression in the larval gut, while those constitutively expressed in all life stages are involved in other life processes. We also compared T. castaneum cysteine peptidase genes to those in other model insects and coleopterans. The combined data has allowed us to propose a model of cysteine peptidase function in T. castaneum.

Methods

Insects

The T. castaneum lab colony originated from a grain bin in Kansas more than 30 years ago, and has been reared continuously at the Center for Grain and Animal Health Research (CGAHR, Agricultural Research Service, United States Department of Agriculture, Manhattan, KS, USA) on a diet of 95% wheat flour and 5% brewer’s yeast at 28°C, 75% R.H., 0L:24D. For this experiment, insects were subcultured from the laboratory colony, and specific life stages were removed as described. Adults were collected at 3–7 days post-eclosion to ensure they were actively feeding and sexually mature. Eggs were sifted out of diet 24–48 h after oviposition and were separated from the fine particulate with a brush. Larvae were collected at a late instar stage that was actively feeding (approximately 14 days post hatch). Pupae were sifted out and sorted to include those with pigmentation in the eye, but not the elytra. All stages were sampled from different generations to provide three independent biological samples per life stage.

Library preparation and sequencing

RNA was collected from each of the three biological replicates of each life stage (10 adults, approximately 500 eggs, 10 larvae, and 10 pupae per replicate). All samples were pulverized in TRIZOL (BulletBlender, Next Advance Inc., Averill Park, NY, USA) at speed 8 for 2 min with RNAse-free ziroconium oxide beads. RNA extraction and purification was with a Zymo mini prep kit (Irvine, CA, USA). To obtain mRNA, DIRECTbeads (Agilient, Santa Clara, CA, USA) were used to select polyA RNA, and libraries were made with the 200 base pair RNA-seq v2 kit (Life Technologies, Grand Island, NY, USA). Samples were sequenced on 318v2 chips on the Ion Torrent Personal Genome Machine (PGM) (Life Technologies). Each run provided approximately 1–5 million reads, with a total of 5–12 million reads per life stage (Table S1).

Data analysis

Data was analyzed using ArrayStar (Lasergene Genomics Suite v12.0.0, DNASTAR, Madison, WI, USA) by mapping reads to the Tcas3 genome build. Read counts were normalized by Reads Per Kilobase of template per Million mapped reads (RPKM Mortazavi et al., 2008). A Kruskal-Wallis test was done among the life stages (df = 3), calculated from linear total RPKM values.

Sequence alignments were made with ClustalW (Thompson, Higgins & Gibson, 1994), and phylogenetic tree construction using Maximum Likelihood analysis with 500 bootstrapping iterations was with MEGA (Tamura et al., 2013). Predicted protein sequences were used to construct the T. castaneum cysteine peptidase gene tree. Nucleotide sequences were used for the model insect phylogenetic tree, comparing sequences from T. castaneum to those from the models Apis mellifera and Drosophila melanogaster. Sequences were obtained via NCBI gene search, using conserved domain ‘peptidase_C1A’ (cd02248) as the search criteria for each model species. Nucleotide sequences also were used for the phylogenetic comparison of coleopteran pest cysteine peptidase genes in T. castaneum, Dendroctonus ponderosae, Diabrotia virgiferi virgiferi and Leptinotarsa decemlineata. The other coleopteran insects did not have fully annotated genomes at the time of this analysis, so to find all genes identified as cysteine peptidases, NCBI nucleotide BLAST was used (Altschul et al., 1990) with the following search strategy. First, T. castaneum LOC659441 was used as the query sequence, and search results were limited to the three coleopteran pests of interest. Second, T. castaneum LOC663234 was used as the query sequence and results were limited as before. The second BLAST did not return any new sequences, indicating we had retrieved all possible gene sequences deposited in NCBI using the first sequence. All trees used T. castaneum LOC655494 (aspartic protease) as a rooted outgroup.

Comparisons between D. melanogaster and T. castaneum gut cysteine peptidase expression were made using FlyAtlas Anatomical Expression Data (FlyAtlas.org; Robinson et al. (2013). Levels of expression were determined using default parameters in FlyAtlas (no expression = 0–9.999 RPKM, low expression = 10–99.999 RPKM, moderate expression = 100–499.999 RPKM, high expression = 500–999.999 RPKM, and very high expression > 999.999 RPKM).

Results and Discussion

Overall cysteine peptidase expression patterns

The relative expression of T. castaneum cysteine peptidase genes in different developmental stages was obtained by high throughput sequencing of the transcriptome of three independent biological replicates obtained from adults, eggs, larvae, and pupae (Table 1 and Table S2). Overall, specific cathepsin L and B transcripts were more abundant in the feeding stages (adult and larvae), and overlapped with those previously identified in a larval gut study (Morris et al., 2009) or in response to dietary inhibitors (Oppert et al., 2010). Both classical cathepsin B peptidases and atypical cathepsin B-like peptidases with a shortened occluding loop were found in T. castaneum (Martynov et al., 2015). The expression patterns of cathepsin B peptidase genes from either classical or atypical groups were similar among the different life stages, but the expression levels of the atypical cathepsin B-like peptidase genes were lower overall than classical cathepsin B genes. Cathepsin F and O peptidase genes were expressed in all life stages, but differed in relative expression levels in pupae (5.3 RPKM vs. 115 RPKM, respectively). Some cysteine cathepsin genes were not expressed in any life stage (or were weakly expressed); cathepsin K was expressed weakly only in adults. Many cathepsin B and B-like genes, one cathepsin L gene, and cathepsin F and O genes were predicted to be lysosomal enzymes in their NCBI entries (Table 1, shaded). We also noted that two cathepsin L genes (LOC659226 and LOC659367) and a cathepsin B-like gene (LOC655148) were no longer supported gene models at NCBI at the time of this publication, but the current and previous data analyses included these genes.

Table 1 Relative expression of cysteine cathepsin genes in four developmental stages (n = 3) of T. castaneum, and the overall change in expression in response to peptidase inhibitors (PIs) in the larval gut. Shaded entries have been annotated as “lysosomal” at NCBI.

Gene ID	Protein ID	Tc annotationa	Chromosome	Larval gut expression (RPKM)b	Gut rankc	Annotationb	Adult expression (RPKM)	Egg expression (RPKM)	Larvae expression (RPKM)	Pupae expression (RPKM)	Response to PIsd	
LOC659441	NP_001164001	11001	10	77,200	82	L	3,844	0.50	55,588	11.2	3-fold increase	
LOC659502	NP_001164314	11000	10	25,900	14	L	6,392	0.03	13,551	1.90	7-fold increase	
LOC659226e	XP_970644	11003	10	42.8	7	L	817	734	878	26.4	None	
LOC659367e	XP_970773	11002	10	35.1	2	L homolog	42.9	0.03	24.0	0.03	None	
LOC659565	XP_970951	10999	10	0.28	2	L	3.82	5.75	0.15	1.54	3-fold decrease	
LOC660368	XP_971698	09365	7	2,390	44	L	489	0.16	1,866	0.03	2-fold increase	
LOC660551	XP_971867	09362	7	1.99	1	L	0.61	0.03	0.42	0.03	None	
LOC660428	XP_971752	09364	7	0.98	1	L	8.29	0.13	0.03	0.03	None	
LOC660669	XP_971975	09448	7	0.02	2	L	7.77	0.10	0.89	0.08	None	
LOC663234 (26-29-p)	NP_001164088	09486	7	1,310	NOC	L	233	324	287	818	None	
LOC660491	NC_007422	pseudogene	7	–	NOC	L	0.03	0.03	0.03	0.03	–	
LOC656198f	XP_967834	09217	7	28.6	1	B homolog	35.9	25.2	44.5	4.04	–	
LOC662417	XP_973607	–	7	2.32	NOC	F	57.9	18.7	42.7	5.30	–	
LOC663145	XP_974298	02952	3	3,140	43	B	669	0.57	1690	1.47	7-fold increase	
LOC663117	NP_001164205	02953	3	1,130	46	B	65.6	0.03	162	4.42	None	
LOC663090	XP_974244	02954	3	249	9	B	93.3	0.11	148	0.54	2-fold decrease	
LOC663066	XP_974220	02955	3	79.9	2	B-like	337	188	372	14.6	2-fold decrease	
LOC658343	XP_969833	02843	3	0.02	1	L	0.36	0.10	0.03	0.03	None	
LOC655148e	XP_966750	05431	8	443.27	3	B-like	112	0.13	59.5	0.03	5-fold decrease	
LOC655077	XP_966663	05432	8	1.18	3	B-like	0.10	0.03	0.03	0.03	4-fold decrease	
LOC657117	XP_968689	05954	8	57.6	NOC	B homolog	9.14	0.03	1.65	0.33	18-fold increase	
LOC657203	XP_968767	05953	8	82.2	3	B-like	4.16	0.03	10.8	0.03	20-fold increase	
LOC656957	XP_008196467	05955/05956	8	–	NOC	B-like	9.87	0.03	0.64	0.08	4-fold increase	
LOC657038	XP_008196465	–	8	–	NOC	B-like	0.03	0.03	0.03	0.03	–	
LOC659087	XP_970512	07214	4	12.0	NOC	O	36.1	35.3	15.4	115	1.3 decrease	
LOC100141668	XP_001814509	13582	1 (X)	0	1	K	1.24	0.03	0.10	0.03	–	
Notes.

a From Tribolium Genome Sequencing Consortium et al. (2008).

b From Martynov et al. (2015).

c As defined in Morris et al. (2009), from microarray gene expression data from larval gut tissue (higher ranks = higher expression); NOC—not on chip.

d PIs, peptidase inhibitors E-64 and STI, values are fold-change in gene expression from the larval gut, from Oppert et al. (2010).

e No longer supported as gene models at NCBI.

f Annotated as tubulointerstitial nephritis antigen-like at NCBI.

Differential expression patterns of the RNA-Seq data were more easily discernible in a heat map of gene expression across the T. castaneum life cycle (Fig. 1). Blocks of color within the heat map highlighted three main patterns: (1) increased expression during adult and larval stages relative to pupae and eggs (yellow to red in contrast to blue to grey in other stages, herein referred to as pattern group one), (2) constitutive expression across all stages (yellow to orange in all stages, herein referred to as pattern group two), and (3) low to no expression across all stages (blue to grey in all stages, herein referred to as pattern group three).

Figure 1 Heat map of the relative expression cysteine peptidase genes across life stages of T. castaneum.

Red and warm tones indicate high expression and blue and cool tones indicate low expression. The gene loci identification is to the right of the heat map (see Table 1 and Table S2 for more information).

Pattern group one included 12 genes encoding cathepsin L or B cysteine peptidases (Fig. 2A). Similar to our previous data from the larval gut (Morris et al., 2009; Martynov et al., 2015), we observed that three cathepsin L genes, LOC659441 and LOC659502 on chromosome 10, and LOC660368 on chromosome 7, were the most highly expressed cysteine peptidase genes among all of the developmental profiles in T. castaneum (Table 1). These genes were primarily expressed in larvae and adults, and the difference in expression across developmental stages was significant (Kruskal-Wallis test, p < 0.05; Table S2). The most highly expressed cathepsin B peptidase gene in adults and larvae was LOC663145 on chromosome 3 (p < 0.03). Other genes in pattern group one with lower expression levels encoded cathepsin L (LOC660669), cathepsin B (LOC663117 and LOC663090), cathepsin B-like (LOC655148, LOC656957 and LOC657203), and L and B homologs (LOC659367 and LOC657117, respectively). Homologs lack conserved residues of typical cathepsins. We propose that members of this pattern group have some role in dietary protein digestion, as they are most abundant during feeding stages.

Figure 2 Line graphs grouping data by three different expression patterns of T. castaneum cysteine peptidase genes.

(A) Increased relative expression in adult and larvae stages. (B) Constitutive expression across all life stages. (C) Low to no expression across all life stages. Y-axis shows logarithmic RPKM values, transformed from raw data. Dotted grey line shows baseline for expression; data points above this level have increased expression and data points below have reduced expression relative to all other genes in the data set.

Pattern group two was comprised of a suite of six genes constitutively expressed across all life stages, albeit at overall lower levels than the previous group (Fig. 2B). Genes displaying this expression pattern included cathepsin L (LOC659226, LOC663234), cathepsin B-like (LOC663066), cathepsin B homolog (LOC656198), cathepsin F (LOC662417), and cathepsin O (LOC659087). Although expressed in all stages, one cathepsin L (LOC663234) and cathepsin O (LOC659087) were expressed much higher in pupae (Table 1). We hypothesized that these constitutively expressed genes are not involved in food digestion but in other life processes, and probably are secreted to the lysosome.

Pattern group three contained eight cysteine peptidase genes with little or no expression across all stages (Fig. 2C). Genes in this group included cathepsin L (LOC659565, LOC660428, LOC660551, LOC658343), cathepsin B-like (LOC655077, LOC657038), cathepsin K (LOC100141668), and a pseudogene (LOC660491). While these genes may be considered “minor” cysteine cathepsins by their expression levels, we did find differential expression among developmental stages for some (LOC659565, LOC660428, LOC100141668), suggesting they may have stage-specific roles. However, some of these genes may be vestigial, evolutionary remnants of cysteine peptidase genes.

Interestingly, genes encoding homologs that lack conservation in critical amino acid residues for cysteine peptidase function displayed differential expression patterns. Cathepsin L-homolog (LOC659367) and cathepsin B-homolog (LOC657117) were expressed at higher levels in adult and larvae and were in pattern group one (Fig. 2A). Cathepsin B-homolog (LOC656198; pattern group two) was expressed in all life stages, although considerably lower in pupae (Fig. 2B and Table 1). While these peptidases may not have a catalytic role, their expression patterns suggest that they have biological relevance in T. castaneum.

Comparison of RNA-Seq to tiling arrays

Tiling array information for developmental stages is available in Beetlebase (http://beetlebase.org/). This information was generated for gene annotation and not for relative expression. However, the tiling array surveyed more time-points in T. castaneum development (6 h, 14 h, and 30 h embryonic; early, mid and late larval; early, mid, and late male and female pupae; early and late male and female adult) and in some cases demonstrated more details about the relative expression profile of some peptidase genes within a developmental stage (Figs. S1–S5). Mostly the RNA-Seq and tiling array profiles were similar, with these exceptions in tiling array profiles compared to RNA-Seq data: cathepsin L LOC660669 was more highly expressed in embryos (Fig. S2); cathepsin B-like LOC655077 had increased expression in all stages (Supplemental Fig. S3A); cathepsin L homolog LOC659367 was expressed less in larvae and adult (Fig. S4); cathepsins L and K, LOC658343 and LOC100141668, respectively, were expressed more in embryos, and cathepsin B homolog LOC656198 was expressed less in adults (Fig. S5). For non-constitutively expressed genes identified in our RNA-Seq study, expression patterns in the tiling array were sometimes cyclical, with increased expression particularly in late embryo, early to mid larval, and throughout adult stages, suggesting the expression was coordinated and regulated (Figs. S1–S4).

Integration of RNA-Seq and iBeetle phenotypes

An RNAi screen at iBeetle (http://ibeetle-base.uni-goettingen.de; Dönitz et al., 2015) documents the phenotypic response of T. castaneum after injection with gene-specific dsRNA at the larval and pupal stage. Of the 25 cysteine cathepsins in our study, eight have been analyzed in iBeetle (Table S3). In general, knockdown of the B, B-like, and B homolog cathepsins (LOC663066, LOC663145, LOC656957, LOC656198, LOC663090, LOC657117) had little phenotypic change from wild type and did not cause significant mortality. An exception, cathepsin B homolog LOC656198, had 20% mortality 22 days post-larval injection, and pupae were unable to eclose.

Injection of dsRNA for cathepsin L LOC659226 induced some mortality post larval injection (20–30%), but interestingly, adults that survived had no living offspring (Table S3). In our study, LOC659226 was constitutively expressed, but lower in pupae, which supports the fact that knockdown during the pupal stage had little or no effect on development or phenotype in iBeetle. LOC659226 was also characterized as pattern group two in this study, and as knockdown in larvae affected fecundity of adults, it may be involved in embryogenesis.

Perhaps the most interesting knockdown was cathepsin L LOC660368, where all beetles injected as larvae died (Table S3). Survivorship after pupal injection was much higher (only 10% mortality), but adults had reduced fecundity due to egg defects. In our study, LOC660368 was highly expressed during larval and adult stages (part of pattern group one) and therefore proposed to be involved in digestion. The data demonstrates that LOC660368 is apparently crucial to beetle survival.

Response of cysteine peptidase gene expression to inhibitors

Previously, we used microarrays to evaluate the larval gut response to dietary inhibitors E-64 (trans-epoxysuccinyl-L-leucylamido(4-guanidino)butane), a cysteine peptidase inhibitor with broad specificity, and STI (soybean trypsin inhibitor) typically a serine peptidase inhibitor (Oppert et al., 2010). In that study, we found that highly expressed cathepsin L and B genes in the larval gut (LOC659441, LOC659502, LOC660368, and LOC663145) were increased in expression even more when larvae were fed diets containing peptidase inhibitors (PIs, summarized in Table 1). This response suggests a protective compensation mechanism to retain the expression of these enzymes in larvae. Cathepsin B-like (LOC656957 and LOC657203) and cathepsin B homolog (LOC657117), expressed at low to moderate levels in adult and larval feeding stages in this study, were highly upregulated (from 4 to 20-fold) when larvae were fed E-64. All of these genes were found in pattern group one of this study, and all of these enzymes are likely important in protecting the insect from the deleterious effects of cysteine peptidase inhibitors through additional compensation mechanisms. Cathepsin L (LOC659565), cathepsin B (LOC663090) and cathepsin B-like (LOC655077, LOC655148, and LOC663066) genes decreased in expression when inhibitors were fed to larvae, even though they were not highly expressed in the control larval gut (Morris et al., 2009) or other life stages (this study). These genes are likely sensitive to the inhibitors and lack a feedback mechanism for upregulation.

Phylogenetic analysis of T. castaneum cysteine cathepsins

We aligned predicted protein sequences (excluding pseudogene LOC660491) and overlaid expression pattern data for a phylogeny of all T. castaneum cysteine peptidases (Fig. 3). Cysteine peptidases clustered into two main clades based on function, cathepsin L and B. In general, most subclades represented either pattern group one (red shading) or pattern group two (yellow shading) with pattern group three genes (blue shading) intercalated within each clade. For example, in clade B, the subclade including LOC657117, LOC657203, LOC656957, and LOC657038 (bottom of the tree) had high sequence similarity, but only three of the four had increased expression during feeding stages. In clade L, highly expressed genes LOC660368 and LOC660669 were similar to LOC658343 (top of tree), which was expressed at lower levels. The genes expressed at low levels may be acquiring loss of function mutations.

Figure 3 Phylogenetic tree of T. castaneum cysteine peptidases, using the predicted protein sequence.

Maximum likelihood tree with 500 iterations of bootstrapping were used; bootstrapping values are represented as a percentage (0–100) by each branch. Genes shaded in red and labeled as pattern group 1 (P1) had increased relative expression in feeding stages; genes shaded in yellow were expressed constitutively in all developmental stages and labeled pattern group 2 (P2); genes shaded in blue had comparatively low to no expression and fell in pattern group 3 (P3). Genes with high (>400 RPKM) expression during feeding stages only are indicated with an asterisk. The chromosome number where each gene is located is shown in parentheses along with the pattern group.

Genes in pattern group one also grouped mostly by chromosomal location. Clusters of cathepsin L peptidases on chromosome 10 and clusters of cathepsin B and B-like peptidases from chromosome 3 and 8 were found within subclades (Fig. 3). In clades B and L, each chromosome group also included at least one gene with low/no expression. This may be evidence of recent gene duplication events, with some genes retaining functions, while others loose or gain new function and thus have reduced expression compared to their digestive counterparts. LOC660491 is annotated in the NCBInr database as a cathepsin L pseudogene and was excluded from the phylogenetic tree. This pseudogene is found on chromosome 7 in a cluster of five cathepsin L genes with varied expression levels, located between LOC660551 and LOC660428, both with low expression in all life stages. However, other cathepsin L genes on chromosome 7 were either highly or constitutively expressed.

Cathepsin F (LOC662417), K (LOC100141668), and O (LOC659087) peptidase genes are singular representatives from each cysteine peptidase type and cluster together in a subclade, although bootstrap values were very weak (bootstrap = 6 and 24). Phylogeny demonstrated that they are more similar to cathepsin L genes. Cathepsin F and O genes are constitutively expressed, suggesting an indispensible function across life stages (Table 1).

Figure 4 Phylogenetic relationship of cysteine peptidases between T. castaneum and model insects, Drosophila melanogaster and Apis mellifera.

Maximum likelihood tree with 500 iterations of bootstrapping. Bootstrapping values are represented as a percentage (0–100) by each branch. Shading represents expression patterns (P1-3) as in Fig. 2 and the letter represents the general cathepsin type. Genes with high (>400 RPKM) expression during feeding stages only are indicated with an asterisk. Tribolium castaneum aspartic protease protein was used as an outgroup. Brackets denote groups with high (99%) bootstrap values. D. melanogaster genes with high to very high expression in the midgut as defined by default parameters in FlyAtlas Anatomical Expression Data larval vs. adult (FlyBase.org) are indicated with H (high) or VH (very high).

Phylogenetic analysis among model insects

A phylogenetic analysis with the model insects Drosophila melanogaster and Apis mellifera showed specific grouping among taxa (Fig. 4). D. melanogaster and A. mellifera have substantially fewer annotated cysteine peptidase genes than T. castaneum, and in several cases a large cluster of highly upregulated T. castaneum genes were lone representatives in a subclade. Highly expressed T. castaneum genes from pattern group one grouped into two subclades, with cathepsin L genes LOC659441 and LOC659502 and L homolog LOC659367 in one subclade (bootstrap = 99), and cathepsin L LOC660368 in another subclade (bootstrap = 89) with genes expressed at lower values. Similarly, highly expressed cathepsin B LOC663145 was in a subclade (bootstrap = 99) with other pattern group one genes, as well as constitutively expressed LOC663066 and low expression genes LOC655077 and LOC667038. Putative digestive cysteine peptidases forming unique subclades points to a relatively recent expansion within the T. castaneum lineage. These gene duplications may have occurred after the split from Hymenoptera and Diptera, approximately 200–300 million years ago (Misof et al., 2014).

We can find only a few D. melanogaster and no A. mellifera studies of cysteine peptidase gene function in the literature. Three D. melanogaster cysteine peptidase genes had increased expression in the fly midgut and hindgut (Cp1, VH-very high expression, and 26-29-p, H-high expression, from the cathepsin L group, and CG12163, VH-very high expression from the cathepsin B group) (Fig. 4), as defined by default parameters in FlyAtlas Anatomical Expression Data in larva and adult (FlyBase.org, dos Santos et al., 2015). Interestingly, the T. castaneum ortholog to Cp1 is the pseudogene LOC660491 (bootstrap = 48). Therefore, we speculate that LOC660491 is ancestral to the duplicated genes on chromosome 7, but has since evolved as a loss of function pseudogene. D. melanogaster Cp1 gene was hypothesized to have a role as a universal housekeeping gene or development because it was expressed in all life stages (Matsumoto et al., 1995). Cp1 was also isolated from a D. melanogaster haemocytic mbn-2 cell line and localized to lysosomes within the cell (Tryselius & Hultmark, 1997). Cp1 has sequence similarity (68%) to lobster cysteine proteinase-3, which was found in lobster digestive “juice” and had significant expression in alimentary organs, suggesting a role in digestion in this species (Laycock et al., 1989). Cp1 grouped with other putative non-digestive genes in T. castaneum, indicating that the beetle orthologs also are probably lysosomal.

In contrast, 26-29-p cathepsin L genes were essentially identical (bootstrap = 99) and highly expressed in the larval gut of both the fly and beetle, as well as in all developmental stages of T. castaneum, suggesting a common function in both species. In D. melanogaster, gene 26-29-p is named for the expressed protein molecular mass (26/29 kDa) and identification as a “proteinase” (p) (Fujimoto et al., 1999). 26-29-p has a role in insect immunity defense mechanisms and is conserved in a number of insects. T. castaneum 26-29-p was continuously expressed across all life stages and thus also could provide a general immune response. D. melanogaster 26-29-p mutants had enlarged rhabdomeres and ommatidia and therefore a potential role in eye development via apoptosis (Gambis et al., 2011). Thus, 26-29-p may have a ubiquitous role degrading proteins in general processes such as immunity and development.

D. melanogaster CG12163 had low similarity to T. castaneum LOC662417 (bootstrap = 26), a putative cathepsin F with constitutive expression across life stages. However, the T. castaneum cathepsin F was nearly identical to A. mellifera LOC408851 (bootstrap = 99), suggesting a similar function in these insects, although empirical data is not available.

Phylogenetic analysis among coleopteran pests

The coleopteran tree revealed a wide range in the number of cysteine peptidases in selected beetle species available at the time of this study (Fig. 5). Dendroctonus ponderosae had a total of 53 cysteine peptidases, Leptinotarsa decemlineata had 36 and Diabrotica virgifera virgifera had eight. The lower count in D. virgifera virgifera may be an artifact, due to the lack of a complete genome sequence. The tree did not group by species, but by peptidase type, mainly type L and B, indicating conservation of these enzyme types in coleopterans. In many cases, large groups of genes from a single species clustered together with high sequence similarity (bootstrap values >85). In these cases, the clade was collapsed for ease of reading, and the number of original sequences is shown in parentheses. T. castaneum genes are highlighted to show gene expression patterns as in Figs. 3 and 4. In general, constitutively expressed genes had orthologs in other beetles, while the highly expressed genes in feeding stages appear to be more divergent. Overall, there was evidence of an expansion/duplication of cathepsin L peptidase genes in D. ponderosae and L. decemlineata, whereas expansion of cathepsin B genes was more evident in T. castaneum and D. ponderosae.

Figure 5 Phylogenetic relationship between T. castaneum and other pest coleopterans, Diabrotica virgifera virgifera, Dendroctonus ponderosae, and Leptinotarsa decemlineata.

The tree was constructed using maximum likelihood with 500 iterations of bootstrapping. Bootstrapping values are represented as a percentage (0–100) by each branch. Coloring represents expression patterns (P1-3) as in Fig. 2. Genes with high (>400 RPKM) expression during feeding stages only are indicated with an asterisk. T. castaneum aspartic protease gene was used as an outgroup. Branches with more than one cysteine peptidase and with high sequence similarity were reduced to a single branch and noted with number of total genes in parentheses.

The phylogenic analyses illustrated that a few cysteine peptidases are conserved across insects, but after coleopterans branched from other insects, cysteine peptidases duplicated many times, presumably to preserve effective food digestion in a hostile environment of plant antidefense compounds. Duplication led to many genes of similar function, which reduced the selective pressure on an individual gene. Increased expression of cysteine peptidase genes or mutations protected these insects from harmful dietary inhibitors and improved dietary efficiency. We found evidence of this in tenebrionids, where the substrate binding regions in cysteine peptidases were highly variable (Martynov et al., 2015). These phenomena allowed for the accumulation of mutations and eventually led to loss of function or new function (neofunctionalization). In T. castaneum, L. decemlineata, and D. ponderosae, cysteine peptidase genes were greatly expanded in gene duplication events. The expansion in the two field pests (L. decemlineata, and D. ponderosae) appears to be more expansive, probably because of the intense selection pressure these insects incur from a more diverse diet.

Figure 6 Venn diagram of the functional characterization of 25 cysteine peptidase genes and associated pseudogene in T. castaneum.

Each circle represents one of three pattern groups, P1, P2, and P3. Genes within the red circle (P1) indicate those most likely involved in digestion, either directly or when inhibitors are present. Genes within the yellow circle (P2) indicate those most likely to be lysosomal or with ubiquitous function. Genes within the blue circle (P3) show low/no expression in our study or with dietary inhibitors. The genes that fall into multiple categories are within the overlapping portions of the circles. Genes that were highly upregulated when T. castaneum was fed digestive inhibitors are marked with an asterisk. The most highly expressed genes are showed in bold. The seven genes with a major role in digestion are underlined.

Development of a functional model for cysteine peptidases in T. castaneum

We synthesized all of the evidence thus far into a working model of cysteine peptidase function in T. castaneum (Fig. 6; an alternate figure containing the GLEAN numbers from the original annotation can be found in Fig. S6). We concluded the 12 genes in pattern group one have higher expression during feeding stages of T. castaneum, and all are potentially involved in food digestion (Fig. 6, genes in red circle not overlapping with other circles). Of those, ten have multiple lines of evidence suggesting involvement in digestion: increased expression during feeding stages (this study), high gut rank in a previous microarray analysis (Morris et al., 2009), high expression in the larval gut in a previous RNA-Seq experiment (Martynov et al., 2015), and in most cases differential expression in response to dietary inhibitors (Oppert et al., 2010; Fig. 6, first ten listed in red circle). These ten include seven cathepsin L and B genes with a major role in digestion, (Fig. 6, underlined) and three with a minor role when fed a normal diet. In a previous biochemical study, we described seven distinct cysteine peptidase activities from fractionation of gut enzymes in the T. castaneum gut, and all were anionic enzymes mostly in the anterior midgut (Vinokurov et al., 2009); we propose that the seven major cysteine cathepsin digestive genes listed here encode those seven anionic enzymes. Three cathepsin B or B-like genes (LOC657117, LOC657203, and LOC656957) and four cathepsin L genes (LOC659441, LOC659502, LOC660368, and LOC663145) were upregulated when T. castaneum larvae were fed digestive inhibitors (Fig. 6; asterisk); these genes provide optimal targets to control this damaging stage of the beetle as they apparently provide a compensation response. Two genes in pattern group three with low expression were further reduced (LOC655077 and LOC659565) when fed inhibitors (Fig. 6; overlap with red and blue). One constitutively expressed gene (LOC663066) also decreased when larvae were fed inhibitors (Fig. 6; overlap with red and yellow).

Genes that were constitutively expressed (LOC659226, LOC663234, LOC658087, LOC662417, and LOC656198; Fig. 6, yellow circle) also are potential targets for control via RNAi; while some of these may be lysosomal, lysosomal enzymes have been demonstrated to be amenable to RNAi (Koo et al., 2008). The cysteine peptidase genes that had low to no abundance in any of the developmental stages evaluated in this study (Fig. 6, blue circle), may be expressed at other developmental times than those sampled in this study, in different environments or diets, or at low levels in specific tissues. Alternatively, some may represent genes acquiring loss-of-function mutations, eventually to become additional pseudogenes.

We also note that comparable expression patterns in both larvae and adults suggest that both stages can be targeted in pest control strategies. LOC659441 and LOC659502 were the most highly expressed among all cysteine peptidase genes in adults and larvae. Expression was higher in larvae than in adults (15-fold and 2-fold increased expression, respectively), suggesting that larvae are more efficient in dietary protein hydrolysis and would be the more economically damaging stage. However, larvae persist approximately 30 days, whereas adults can live up to two years, and they may be more easily targeted with biopesticides. Additionally, targeting individual genes that are part of a group with similar function and compensation mechanism may not be effective. Therefore, more unique cysteine cathepsins might be better genetic targets. The phenotypic data from the iBeetle RNAi screen indicated that targeting LOC6630368 and LOC659226 at the larval stage resulted in no living offspring, suggesting either gene could be an excellent candidate for oral RNAi pest control methods.

Conclusions

We found that developmental expression profiles provided clues to the function of individual cysteine peptidase genes in T. castaneum. When we combined the data with previous studies as well as phylogenetic studies, we found considerable evidence that coleopteran cysteine peptidase genes have duplicated and evolved in response to selection pressure, and are continuing to do so for improved digestion. This information will be useful in understanding and characterizing the expanded cysteine peptidase gene family in T. castaneum and in other beetles. A greater understanding of digestion and peptidase activity across the life cycle of this pest species will be useful in the development of new pest control products and strategies.

Supplemental Information

Figure S1 Tiling arrays for cysteine cathepsin genes found on chromosome 3 (beetlebase.org)

Data extracted was from: 6 h, 14 h, and 30 h embryonic; early, mid and late larval; early, mid, and late male and female pupal; early and late male and female adult.

Click here for additional data file.

Figure S2 Tiling arrays for cysteine cathepsin genes found on chromosome 7 (beetlebase.org)

Data extracted was from: 6 h, 14 h, and 30 h embryonic; early, mid and late larval; early, mid, and late male and female pupal; early and late male and female adult.

Click here for additional data file.

Figure S3 Tiling arrays for cysteine cathepsin genes found on chromosome 8 (a and b are different clusters; beetlebase.org)

Data extracted was from: 6 h, 14 h, and 30 h embryonic; early, mid and late larval; early, mid, and late male and female pupal; early and late male and female adult.

Click here for additional data file.

Figure S4 Tiling arrays for cysteine cathepsin genes found on chromosome 10 (beetlebase.org)

Data extracted was from: 6 h, 14 h, and 30 h embryonic; early, mid and late larval; early, mid, and late male and female pupal; early and late male and female adult.

Click here for additional data file.

Figure S5 Tiling arrays for cysteine cathepsin genes found on different chromosomes (beetlebase.org)

Data extracted was from: 6 h, 14 h, and 30 h embryonic; early, mid and late larval; early, mid, and late male and female pupal; early and late male and female adult.

Click here for additional data file.

Figure S6 Alternative model

An alternate model containing TC gene numbers (i.e., GLEAN numbers from the early annotation project), which some Tribolium researchers may find easier to use. See Fig. 6 for corresponding LOC gene numbers.

Click here for additional data file.

Table S1 Sequencing run data

General information on sequencing runs for each biological replicate of each life stage. Total reads refers to the number of useable reads.

Click here for additional data file.

Table S2 RPKM and statistic values

The mean linear total RPKM values for each cysteine peptidase gene in each T. castaneum life stage with standard error for each mean. P-value is from an Analysis of Variance (ANOVA) of expression values (RPKM) across life stages after multiple testing correction (FDR) and before correction (none).

Click here for additional data file.

Table S3 iBeetle phenotypes

Data is the percent of individuals with the defined mortality/morphology out of 10 individuals. Blank boxes indicate information was not available. Primers are those used in the RNAi screen (http://ibeetle-base.uni-goettingen.de; Dönitz et al., 2015).

Click here for additional data file.

We would like to thank Tom Morgan for help maintaining and collecting insects. Mention of trade names or commercial products in this publication is solely for the purpose of providing specific information and does not imply recommendation or endorsement by the US Department of Agriculture. USDA is an equal opportunity provider and employer.

Additional Information and Declarations

Competing Interests

Author Contributions

DNA Deposition

Data Availability

Brenda Oppert is an Academic Editor for PeerJ.

Lindsey Perkin conceived and designed the experiments, performed the experiments, analyzed the data, wrote the paper, prepared figures and/or tables, reviewed drafts of the paper.

Elena N. Elpidina and Brenda Oppert conceived and designed the experiments, contributed reagents/materials/analysis tools, reviewed drafts of the paper.

The following information was supplied regarding the deposition of DNA sequences:

NCBI SRA BioProject accession SRP065255.

The following information was supplied regarding data availability:

NCBI SRA BioProject accession SRP065255.

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
