# Peer review of "Expression patterns of cysteine peptidase genes across the Tribolium castaneum life cycle provide clues to biological function"

_PeerJ, doi:10.7717/peerj.1581_

## Round 0.1 · original submission · Major Revisions

Dear Dr. Perkin,

Please address Reviewer 1's comments on the reproducibility of the results and Reviewer 2's comments on the statistical procedures. Regarding the reproducibility of the results, if the generation of data replicates is too costly and not feasible, please consider the use of statistical perturbation methods (e.g. bootstrap) to evaluate the uncertainty of the discoveries.

Reviewer 1 ·

Basic reporting

No Comments

Experimental design

1. Need replicate samples
The samples in Figure 1 and Figure 2 do not have replicates for the same biological condition. It is unclear whether the observed patterns in Figure 1 and Figure 2 are robust for biological noises. I suggest the authors to add more replicate samples in their experiment.

2. Gene expression patterns
The manuscript is lack of experimental confirmation of the gene expression patterns (e.g. qRT-PCR). The distinct gene expression patterns are central to this study. Although the authors compared their results with tiling array data, more solid experimental validations are necessary.

Validity of the findings

1. Phylogenetic analysis within Tribolium castaneum
In page 12 Line 263, the author claimed that “To determine if sequence similarity can shed light on gene function, we aligned predicted protein sequences (excluding pseudogene LOC660491) and overlaid expression pattern data”. However, throughout this paragraph, the authors merely described the phylogenetic patterns and their expression patterns, and did not discuss the linkage between the biological functions of the genes and their sequence similarity. The authors may interpret the patterns they observed.

2. Phylogenetic analysis with other insect species
The authors may rewrite this section, as the current version is hard to follow. This section is lengthy. I suggest the authors may split it into two parts.

Page 14 Line 303-305: “The data suggests that gene duplication occurred within the T. castaneum lineage after the split from other insects, such as honeybee and fruit fly.” This is an important conclusion in this manuscript. However, I do not think that the description from this paragraph can be directly drawn to this conclusion. The authors may provide more evidence that the cysteine peptidase genes duplicated in T. castaneum lineage.

Additional comments

MINOR ISSUES

1. The resolution of figure 4 is too low. It is difficult to see the species’ names.

2. Figure 1 lacks color scale.

3. Figure 4-6 lack color legends.

4. Page 14 Line 298/302 “pattern group one”, “with other pattern one genes”
Please make these terms more clear.

5. Page 14 Lines 308-312
“Three D. melanogaster cysteine peptidase genes had increased expression in the fly midgut and hindgut (Cp1, VH-very high expression, and 26-29-p, H- high expression, from the cathepsin L group, and CG12163, VH-very high expression from the cathepsin B group) (Fig. 4), as defined by default parameters in FlyAtlas Anatomical Expression Data in larva and adult (FlyBase.org).” Please cite the reference of FlyBase.

6. Conclusions
“We found considerable evidence that coleopteran cysteine peptidase genes have duplicated and evolved in response to selection pressure, and likely are continuing to do so”. The basis for this statement is not clear.

Reviewer 2 ·

Basic reporting

No Comments

Experimental design

No Comments

Validity of the findings

No Comments

Additional comments

In this manuscript, the authors compared the gene expression profiles of 26 cysteine peptidase genes in Tribolium castaneum in 4 developmental stages, and compared these genes with those in other model insects and coleopteran pests. They predicted key cysteine peptidases which were likely involved in food digestion. Overall, the manuscript is well written, but there are also a few questions:

ANOVA was performed to test if there were any differences in RPKM across 4 developmental stages. However, it seems to me that only 3 data points were used in each stage, therefore it is difficult to check the normality assumption required for the validity of ANOVA. Nonparametric tests which are robust to violation of the normality assumption, such as the Kruskal-Wallis test, might be more appropriate.

It might be better to present both the naïve p-values and FDR adjusted p-values in Table S2. There are several genes with p-value 0.00, likely due to rounding. It is unclear whether rounding was performed before or after the Benjamini-Hochberg procedure, which could yield different results. Using the scientific notation should be more informative.

---

## Round 0.2 · Minor Revisions

Please address both reviewers' comments on the revised version, especially those from Reviewer 2 on the validity and presentation of the statistical results.

Reviewer 1 ·

Basic reporting

No Comments

Experimental design

No Comments

Validity of the findings

No Comments

Additional comments

The word size in Figure 1, Figure 4 and Figure 5 is too small. The authors should make them larger.

Reviewer 2 ·

Basic reporting

No Comments

Experimental design

No Comments

Validity of the findings

No Comments

Additional comments

The authors have revised their manuscript, but I still have concerns about the validity of ANOVA in Table S2. The authors stated in the response letter that “the ANOVA analysis was not critical to the interpretation of the data, but we included it as additional information”. As long as the authors decide to include these results, they should be aware that inappropriate use of ANOVA could lead to misleading results and may harm the field in the long run (future researchers in the same field may follow them and simply use ANOVA without checking the normality assumption, which could lead to statistical malpractice and invalid p-values). As I mentioned in my previous comments, nonparametric tests such as the Kruskal-Wallis test would be more appropriate for such a small sample size since it is difficult to check the normality assumption (only 3 data points in each stage). Alternatively, the authors can choose to remove this table to avoid confusion if they believe these results are not critical.
Rounding p-values to 0.00 is generally not a good idea, no matter whether they are naïve p-values or FDR adjusted p-values (e.g., there is a big difference between 0.004 and 0.00000001). The scientific notation should be more appropriate (instead of 0.00) for p-values less than 0.01.
Minor issue: the p-values should be human-readable. They should look like 4.8 × 10-3 (where "-3" is in the superscript) instead of 4.8e-3. They should also have the same number of digits: 3.64e-3 and 3.63e-7 should be rounded to 3.6 × 10-3 (where "-3" is in the superscript) and 3.6 × 10-7 (where "-7" is in the superscript), respectively.

---

## Round 0.3 · accepted · Accept

This revised version has properly addressed Reviewer 2's comments on the statistical analyses.

Reviewer 2 ·

Basic reporting

No Comments

Experimental design

No Comments

Validity of the findings

No Comments

Additional comments

The authors have appropriately addressed my concerns.